# Diffusion Networks with Task-Specific Noise Control for Radiology Report Generation

## ABSTRACT

Existing radiology report generation (RRG) studies mostly adopt autoregressive (AR) approaches to produce textual descriptions token-by-token for specific clinical radiographs, where they are susceptible to error propagation problems if irrelevant contents are half-way generated, leading to potential ill-presenting of precise diagnoses, especially when there exist complicated abnormalities in radiographs. Although the non-AR paradigm, e.g., diffusion model, provides an alternative solution to tackle the problem from AR by generating all contents in parallel, the mechanism of using Gaussian noise in existing diffusion models still has significant room to improve when such models are used in particular circumstances, i.e., providing proper guidance in controlling noises in the diffusive process to ensure precise report generation. In this paper, we propose to conduct RRG with diffusion networks by controlling the noise with task-specific features, which leverages irrelevant visual and textual information as noise rather than the stochastic Gaussian noise, and allows the diffusion networks to filter particular information through iterative denoising, thus performing a precise and controlled report generation process. Experiments on IU X-Ray and MIMIC-CXR demonstrate the superiority of our approach compared to strong baselines and state-of-the-art solutions. Human evaluation and noise type analysis show that comprehensive noise control greatly helps diffusion networks to refine the generation of global and local report contents.[1]

## CCS CONCEPTS

• **Computing methodologies** → **Computer vision**; **Natural language generation**.

## KEYWORDS

Radiology Report Generation, Diffusion Networks, Noise Control, Task-specific Noise

## 1 INTRODUCTION

Medical imaging holds a crucial position in clinical medicine and treatment guidance, where physicians are always required to write medical reports based on the syndromes depicted in images and thus create comprehensive professional records for patient references and later processes. As a particular category of medical

[1]Code will be released in the final version of the paper.

images, radiographs play a vital role in assessing patients' health by examining the internal structures of their bodies, and have been widely used in cardiology, dentistry, and pulmonology, etc. Generally, writing reports is a time-consuming job and often error-prone for inexperienced radiologists, which thus drives a series of work [3, 16, 20, 39, 45] on generating reports automatically and precisely. These studies achieve significant success on this topic, proving the feasibility of this research direction.

To effectively generate radiology reports, most existing studies [4, 13, 20, 28, 35, 36, 50, 52, 53] leverage autoregressive (AR) models (e.g., LSTM [11] and Transformer [46]) as their foundation architecture with the encoder-decoder pipeline. In doing so, visual encoders are jointly optimized with text decoders to capture essential semantics from radiograph inputs so as to establish well image-text mapping for the generation process. They normally adopt latent representations to store the semantic information for such mapping and there is a potential deficiency where those representations have ambiguities in conveying all essential abnormalities in the radiograph. The text decoder is thus disrupted by representation noises and has difficulties in generating comprehensive reports. Moreover, the AR-based text decoder has its own problem in susceptibility to error propagation, thus potentially generates contextually incoherent diagnoses if irrelevant contents are half-way produced.

With the recent advances of non-AR paradigm, e.g., diffusion model [9] on text generation [7, 24] and other cross-modal scenarios [2], it thus provides an alternative solution to existing AR-based approaches for RRG. However, in applying diffusion models, it is difficult to perform precise RRG with directly using stochastic Gaussian noises, thus it still requires certain task-specific guidance to help handle the necessary information and smooth the generation process. Although some studies [16, 28, 34, 53] have shown effectiveness by leveraging external medical knowledge as guidance for AR-based approaches, their designs are not easy to be applied to diffusion networks, whose integrity of information optimization is likely to be corrupted. Therefore, an effective approach is expected to enhance diffusion networks for RRG.

In this paper, we propose a non-AR solution for RRG with diffusion networks, namely, ControlDiff, by employing a novel task-specific noise control mechanism to appropriately operate essential cross-modal information in noising and denoising processes. In our approach, we distinguish useful features, e.g., visual representations of different radiographs and textual contents in reports, from others and leverage the non-useful information as the noise in our diffusion networks, rather than the standard stochastic Gaussian one. Particularly, for each modality, we process both global and local information from them to construct our noise vectors, where removing global noises improves the coherence of final reports and removing local ones enhances the accuracy of describing specific regions in input radiographs. Experimental results on two benchmark datasets, i.e., IU X-Ray and MIMIC-CXR, demonstrate the

superiority of our approach against state-of-the-art studies. Human evaluation and quantitative noise type analysis illustrate that choosing different noises affects content filtering during iterative generation, where controlling global noises ensures the overall consistency of the generated reports, and controlling local noise provides fine-grained task-specific guidance for diffusion networks to produce precise reports.

## 2  RELATED WORK

### 2.1  Radiology Report Generation

RRG is a domain-specific extension to image description generation [32]. This task requires to automatically generate reports in the medical domain. To perform the task, existing studies generally follow the encoding-decoding paradigm. They leverage a visual encoder to capture visual features and use a text decoder to produce reports, where advanced architectures such as convolutional neural networks and Transformers are used. To improve the performance of RRG, there are studies that try to identify essential visual and textual features that contribute to the task and leverage them accordingly. These studies leverage regional visual features [22, 45], medical terms [18, 53, 54], knowledge graphs [13, 28, 56], and report templates [20, 25] to generate high-quality reports. There are other studies that put emphasis on improving the cross-modal alignment through attention mechanisms [16, 29, 57], memory networks [3, 39, 48], expert tokens [50], etc. These studies achieve promising performance for RRG, while they mainly rely on the AR paradigm and thus suffer from error propagation issues. Compared with these studies, the approach proposed in this paper is based on diffusion networks, which generate tokens at the same time and thus avoid error propagation issues.

### 2.2  Diffusion Networks

Diffusion models [9] are non-auto-regressive approaches that are widely used for image generation. Recently, the diffusion models and their variants have been applied to text generation tasks and demonstrated as an alternative solution with outstanding performance on cross-modal content generation [19, 42, 43]. Owing to the discrete nature of texts, several studies that use diffusion networks propose to model discrete data with continuous forms, e.g., embedding [7, 24] and bit representations [2, 31]. For example, Li et al. [24] propose diffusion-based networks for text generation by projecting discrete tokens into continuous vectors. Chen et al. [2] model discrete texts with binary bit representations and enhance the text generation process with a self-condition mechanism for image captioning. These studies generally propose new model architecture or utilize new features to improve the denoising process of diffusion models, so as to generate high-quality reports, where the standard stochastic Gaussian noise is used to in the noising process. Compared to these aforementioned studies, our approach utilizes task-specific noise rather than stochastic Gaussian noise to control diffusion networks for RRG.

## 3  THE APPROACH

Given an input radiograph $\mathcal{V}$, our approach generates its corresponding radiology report $\widehat{\mathcal{R}}$ following the pipeline shown in Figure 1 with three main components, namely, the visual encoder, the task-specific noise generator (TNG), and the diffusion networks (DN). Specifically, the visual encoder $f_{VE}$ encodes the input radiograph $\mathcal{V}$ into visual representations $\mathbf{v}$. The TNG $f_{TNG}$ provides a noise vector $\mathbf{n}$ for the diffusion networks $f_{DN}$ through two main components, i.e., the global noise generator (GNG) and the local noise generator (LNG). GNG uses background Visual features shared by most radiographs and non-informative n-grams in the reports to construct the global noise vector $\mathbf{n}_G$; LNC leverages regional visual features of undetected regions in $\mathcal{V}$ and irrelevant medical terms of reports to construct the local noise vector $\mathbf{n}_L$. Then, GNC and LNC fuse $\mathbf{n}^G$ and $\mathbf{n}^L$ into the noise vector $\mathbf{n}$ that is processed by $f_{DN}$ afterwards. Finally, DN utilizes $\mathbf{v}$ and $\mathbf{n}$ to produce $\widehat{\mathcal{R}}$. In the following texts, we illustrate the details of each aforementioned component according to the pipeline sequence.

### 3.1  Visual Encoder

The visual encoder aims to encode the input radiograph $\mathcal{V}$ into a latent representation $\mathbf{v}$. It contains two components, namely, a visual feature extractor $f_{VE}$ and a feature encoder $f_{FE}$. Specifically, $f_{VE}$ is a pre-trained vision backbone model (i.e., ResNet-101 [8]), and $f_{FE}$ follows the standard architecture of Transformer [46] encoder. We first adopt $f_{VE}$ to extract visual features $\mathbf{h}^v$ from $\mathcal{V}$ and obtain $\mathbf{h}^v$ from the last convolutional layer of $f_{VE}$ through

$$\mathbf{h}^v = f_{VE}\left(\mathcal{I}\right) \qquad (1)$$

Then, we employ $f_{FE}$ to encode $\mathbf{h}^v$ into the visual representations $\mathbf{v}$ of $\mathcal{V}$ through

$$\mathbf{v} = f_{FE}\left(\mathbf{h}^v\right) \qquad (2)$$

where $\mathbf{v}$ is used in TNG to produce noise vectors and DN for report generation.

### 3.2  Task-specific Noise Generator

Generally, diffusion networks utilize the stochastic Gaussian noise [2, 10, 23, 55] and present promising results in text generation tasks. Since the noise is not relevant to any task information, it is intuitive to explore whether task-related noise is able to improve model performance. Particularly for RRG, the forward noising process is analogized to adding non-essential information that is not relevant to the abnormalities highlighted in the gold standard report $\mathcal{R}^*$; the denoising process is to eliminate such information from a noise text to reproduce $\mathcal{R}^*$. We propose to control the noise of diffusion networks with task-specific characteristics. Specifically, we consider two types of noise, namely global noise and local noise. We propose global noise generator (GNG) and local noise generator (LNG) in leveraging global and local information from radiographs and reports to provide the two types of task-specific noise for diffusion networks. Details of GNG and LNG are illustrated as follows.

*Global Noise Generator.* The GNC constructs the global noise vector $\mathbf{n}_G$ according to the visual and text global task-specific noise information that is shared by most radiographs and reports. Specifically, for global visual noise information, it is intuitive to regard the background features shared by the most radiographs as the noise. We run an off-the-shelf background segmentation toolkit (e.g., OpenCV [5]) to produce the background feature vector based

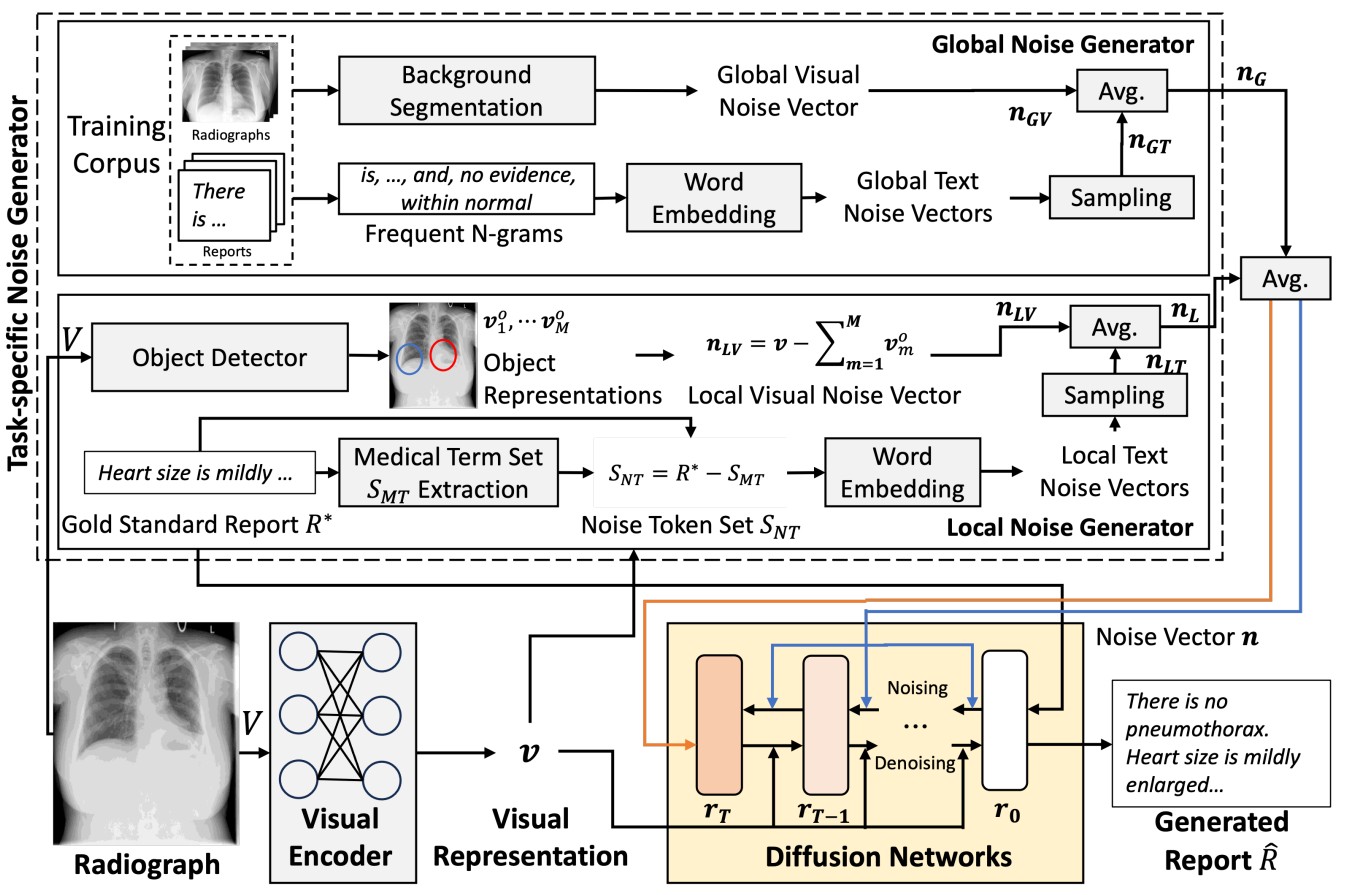

**Figure 1: The overview architecture of our approach for RRG. It consists of three main components, namely, the visual encoder, the task-specific noise generator, and the diffusion networks, which are presented at the left-bottom, top, and right-bottom of the figure, respectively. The blue and orange arrows illustrate how the noise vector n is used in training and inference, respectively. We present an example radiograph for better demonstration.**

on the radiographs in the entire training set and regard it as the global visual noise vector $\mathbf{n}_{GV}$.

For global text noise information, we utilize n-grams that frequently appear in most reports, since they could be interpreted as stop words or report templates that carry limited information about the abnormalities in the radiograph. Specifically, we compute the frequencies of used n-grams in all radiology reports from the training set and select the top-$N_{GT}$ ones as report templates. Then, we map the n-grams into their embeddings and regard them as the global text noise vectors $\mathbf{n}_{GT,1} \ldots \mathbf{n}_{GT,N_{GT}}$. We randomly sample a vector from $\mathbf{n}_{GT,1} \ldots \mathbf{n}_{GT,N_{GT}}$ and regard it as the global ext noise vector $\mathbf{n}_{GT}$.

Finally, the global noise vector $\mathbf{n}^g$ is obtained with the average and normalization (*Norm*) of $\mathbf{n}^{gv}$ and $\mathbf{n}^{gt}$ through

$$\mathbf{n}_G = Norm\left(\frac{1}{2}(\mathbf{n}_{GV} + \mathbf{n}_{GT})\right) \tag{3}$$

*Local Noise Generator.* The LNC constructs the local noise vector $\mathbf{n}_L$ with fine-grained information in the radiographs $\mathcal{V}$ and the gold standard radiology reports $\mathcal{R}^*$. We consider two types of noise from the visual and textual perspectives. For the visual noise, we use

irrelevant regional information in $\mathcal{V}$ to construct the local visual noise vector $\mathbf{n}_{LV}$. Specifically, we employ an off-the-shelf object detector (i.e., fine-tuned Fast R-CNN [40] on Chest ImaGenome [51]) to extract regional visual features $\mathbf{v}^r$ from $\mathcal{V}$. We decompose $\mathbf{v}^r$ into $\{\mathbf{v}_1^r \ldots \mathbf{v}_M^r\}$ along the channel dimension with $M$ as the number of resulting representation. Considering the detected objects are generally essential regions in the radiograph, we use the overall representation obtained from Eq. (2) to subtract the regional features to compute the local visual noise vector $\mathbf{n}_{LV}$.

$$\mathbf{n}_{LV} = \mathbf{v} - \sum_{m=1}^{M}\left(\mathbf{v}_m^r\right) \tag{4}$$

For the text noise, we notice that radiology reports generally contain medical terms (e.g., *heart*, *lungs*, etc.) that are essential for analyzing the diseases of the patients. This motivates us to use non-medical-term words in $\mathcal{R}^*$ to construct the local textual noise vector $\mathbf{n}_{LT}$. Specifically, we firstly train a Transformer-based model to annotate medical terms and use it to extract the medical term set $S_{MT}$ from the gold standard radiology report $\mathcal{R}^*$. Then, we regard the tokens in $\mathcal{R}^*$ yet not in $S_{MT}$ as the noise tokens. Similar to the

process to obtain global text noise, we use the same approach to map the noise tokens into embeddings, randomly sample one from the noise token embeddings, and use it as the local text noise vector $\mathbf{n}_{GT}$. Similar to the process in GNG, we compute $\mathbf{n}_L$ according to $\mathbf{n}_{LV}$ and $\mathbf{n}_{LT}$ through

$$\mathbf{n}_L = Norm\left(\mathbf{n}_{LV} + \mathbf{n}_{LT}\right) \tag{5}$$

Once $\mathbf{n}_G$ and $\mathbf{n}_L$ are obtained, we compute the noise vector $\mathbf{n}$ for the diffusion networks in the following processes by

$$\mathbf{n} = Norm\left(\mathbf{n}_G + \mathbf{n}_L\right) \tag{6}$$

## 3.3 Diffusion Networks

The DN ($f_{DN}$) aims to generate the final report $\widehat{\mathcal{R}}$ based on $\mathbf{v}$ and $\mathbf{n}$. It consists of the diffusion noising and denoising processes, where both processes are used in training, and only the denoising process is used in inference. The following text illustrates the details of training and inference.

*Training.* In training, diffusion noising firstly adds the noise vector $\mathbf{n}$ from TNG into the representation $\mathbf{r}_0$ of the gold standard report $\mathcal{R}^*$ and obtain the noisy representations $\mathbf{r}_t$ at the $t$-th step. The time step $t$ is randomly sampled from a uniformed distribution $U(0, T)$ with $T$ denoting the total number of steps. We follow the approach in DDCap [58] to convert tokens of $\mathcal{R}^*$ into the one-hot representation and compute the representation $\mathbf{r}_t$ at the $t$-th step with $\mathbf{n}$ through

$$\mathbf{r}_t = \sqrt{\bar{\alpha}_t} \cdot \mathbf{r}_0 + \sqrt{1 - \bar{\alpha}_t} \cdot \mathbf{n} \tag{7}$$

where $\bar{\alpha}_t$ is a blending scalar correlated to the noise scheduling strategy of denoising diffusion probabilistic model (DDPM) [9]. Then, $f_{DN}$ reconstructs $\mathbf{r}_t$ to $\mathbf{r}_0$ based on $\mathbf{v}$, where we compute the loss $\mathcal{L}$ through

$$\mathcal{L} = \mathbb{E}_{t \sim U(0,T)} \| f_{DN}(\mathbf{r}_t, \mathbf{v}, t) - \mathbf{r}_0 \|_2^2 \tag{8}$$

The trainable parameters in the model are updated accordingly through gradient descent.

*Inference.* Diffusion denoising generates $\widehat{\mathcal{R}}$ following the standard process of DDCap. It is worth noting that the process to obtain local text noise vector $\mathbf{n}_{LN}$ relies on the gold standard radiology report $\mathcal{R}^*$, which is not available during inference. To handle this issue, we firstly collect a set $\mathcal{S}_{All}$ with all tokens and a set $\mathcal{S}_{MT}$ with all medical terms in the reports of the training data. Next, we randomly sample noise tokens from the difference of $\mathcal{S}_{All}$ and $\mathcal{S}_{MT}$ (i.e., $\mathcal{S}_{All} - \mathcal{S}_{MT}$) and following the same process in training to get the local text noise vector. Then, we denoise $\mathbf{n}$ into the final representation $\widehat{\mathbf{r}}_0$. We initialize $\widehat{\mathbf{r}}_T$ with $\mathbf{n}$ and iteratively subtract noises from $\widehat{\mathbf{y}}_T$ through

$$\widehat{\mathbf{r}}_{t-1} = \sqrt{\bar{\alpha}_{t-1}} \cdot \frac{\widehat{\mathbf{r}}_t - \sqrt{1 - \bar{\alpha}_t} \cdot f_{DN}(\widehat{\mathbf{r}}_t, \mathbf{v}, t)}{\sqrt{\bar{\alpha}_t}}$$
$$+ \sqrt{1 - \bar{\alpha}_{t-1}} \cdot f_{DN}(\widehat{\mathbf{r}}_t, \mathbf{v}, t) \tag{9}$$

Finally, we convert the one-hot representation $\widehat{\mathbf{r}}_0$ into tokens according to the vocabulary and obtain the final radiology report, which is denoted as $\widehat{\mathcal{R}}$.

**Table 1: The statistics of IU X-Ray and MIMIC-CXR, where the numbers of images, reports, patients, and the token-based averaged length (Avg. Len.) of reports in training, validation, and test sets are presented.**

| Dataset | IU X-Ray | | | MIMIC-CXR | | |
|---|---|---|---|---|---|---|
| | Train | Val | Test | Train | Val | Test |
| Image | 5.2K | 0.7K | 1.5K | 369.0K | 3.0K | 5.2K |
| Report | 2.8K | 0.4K | 0.8K | 222.8K | 1.8K | 3.3K |
| Patient | 2.8K | 0.4K | 0.8K | 64.6K | 0.5K | 0.3K |
| Avg. Len. | 37.6 | 36.8 | 33.6 | 53.0 | 53.1 | 66.4 |

## 4 EXPERIMENT

### 4.1 Experiment Settings

**Datasets.** We conduct our experiments on two conventional benchmark datasets, i.e., IU X-Ray [6] from Indiana University and MIMIC-CXR [17] from the Beth Israel Deaconess Medical Center. Table 1 reports the statistics of all datasets in terms of the numbers of radiographs, reports, patients, and word-based report length according to each split of the datasets. Specifically, IU X-Ray is relatively small with 7,470 chest X-Ray images and 3,955 radiology reports. MIMIC-CXR is the largest public radiology dataset with 473,057 chest X-ray images and 206,563 reports. We follow the convention of previous studies [3, 4, 13, 16, 39] by only preserving the "Findings" sections for both datasets. We use the dataset split with the ratio of 7:1:2 in Jing et al. [16] for IU X-Ray and the official split of MIMIC-CXR.

**Baselines.** To evaluate our proposed approach, we compare it with two baseline models in our experiments, including "Trans" and "Diff". "Trans" represents the autoregressive baseline model with ResNet-101 [8] and a three-layer Transformer encoder as the visual encoder, and a three-layer Transformer decoder with an additional eight-head cross-attention layer as the decoder. "Diff" is our baseline model with diffusion networks, which follows the same architecture as DDCap [58] and leverages stochastic Gaussian noise to control the diffusion networks. "Diff+TNG" represents the model where "Diff" is equipped with the proposed task-specific noise control in our approach, denoting our full model.

**Evaluation.** For evaluation metrics, we follow existing RRG studies [3, 4, 13, 21, 39] to evaluate the generated reports with two types of metrics, namely, natural language generation (NLG) and clinical efficacy (CE) metrics. NLG metrics measure the quality of generated reports based on n-gram overlapping, consisting of BLEU [38], METEOR [33] and ROUGE-L [26]. CE metrics evaluate the accuracy of estimating specific medical observations based on the following procedures. First, we adopt CheXpert [14][2] to extract medical labels from both generated and gold standard reports. Then, we calculate the precision, recall, and F1 scores between the

---

[2]CheXpert annotates 14 categories of terms related to thoracic diseases and support devices, including *atelectasis, cardiomegaly, consolidation, edema, enlarged cardiom, fracture, lung lesion, lung opacity, no finding, pneumonia, pneumothorax, pleural effusion, pleural other,* and *support devices.*

**Table 2: The guideline for human evaluation of the reports.**

| METRIC | SCORES | ILLUSTRATION |
|---|---|---|
| FLUENCY | 1 | The report is ungrammatical and hard to understand. |
| | 2 | The report has some grammatical issues but it is understandable. |
| | 3 | The report is grammatical and understandable. |
| COMPLETENESS | 1 | The report misses more than two essential abnormalities in the radiograph. |
| | 2 | The report misses one or two essential abnormalities in the radiograph. |
| | 3 | The report covers all essential abnormalities in the radiograph. |
| PRECISION | 1 | The report contains more than two incorrect descriptions of the abnormalities. |
| | 2 | The report contains one or two incorrect descriptions of the abnormalities. |
| | 3 | The descriptions of the abnormalities in the report are all correct. |

**Table 3: Performance (i.e., the average and standard deviation of three runs with different random seeds) of baselines and our approach (i.e., "+TNG") on the test sets of IU X-Ray and MIMIC-CXR datasets in terms of NLG and CE metrics. We report both the average and standard deviation of three runs with different random seeds. BL-1, BL-2, BL-3, and BL-4 denote BLEU scores using uni-gram, bi-gram, tri-gram, and 4-grams; MTR and RG-L denote METEOR and ROUGE-L, respectively. The average improvement over all NLG metrics compared to "TRANS" is also presented in the "AVG. Δ" column. The relative improvements of our approach over baselines are statistically significant at $p \leq 0.05$ level.**

| DATA | MODEL | NLG METRICS | | | | | | | CE METRICS | | |
|---|---|---|---|---|---|---|---|---|---|---|---|
| | | BL-1 | BL-2 | BL-3 | BL-4 | MTR | RG-L | AVG. Δ | P | R | F1 |
| IU X-RAY | TRANS | $0.385_{\pm0.003}$ | $0.219_{\pm0.002}$ | $0.150_{\pm0.002}$ | $0.105_{\pm0.003}$ | $0.150_{\pm0.002}$ | $0.305_{\pm0.003}$ | - | - | - | - |
| | DIFF | $0.414_{\pm0.004}$ | $0.245_{\pm0.003}$ | $0.162_{\pm0.004}$ | $0.109_{\pm0.002}$ | $0.162_{\pm0.003}$ | $0.312_{\pm0.002}$ | 6.9% | - | - | - |
| | +TNG | $\mathbf{0.508}_{\pm0.003}$ | $\mathbf{0.332}_{\pm0.002}$ | $\mathbf{0.243}_{\pm0.003}$ | $\mathbf{0.189}_{\pm0.002}$ | $\mathbf{0.207}_{\pm0.002}$ | $\mathbf{0.390}_{\pm0.003}$ | 48.6% | - | - | - |
| MIMIC-CXR | TRANS | $0.355_{\pm0.003}$ | $0.213_{\pm0.004}$ | $0.138_{\pm0.002}$ | $0.088_{\pm0.003}$ | $0.126_{\pm0.003}$ | $0.269_{\pm0.002}$ | - | $0.348_{\pm0.002}$ | $0.314_{\pm0.003}$ | $0.330_{\pm0.002}$ |
| | DIFF | $0.373_{\pm0.002}$ | $0.217_{\pm0.003}$ | $0.142_{\pm0.002}$ | $0.101_{\pm0.004}$ | $0.134_{\pm0.002}$ | $0.274_{\pm0.003}$ | 5.5% | $0.385_{\pm0.003}$ | $0.401_{\pm0.004}$ | $0.393_{\pm0.003}$ |
| | +TNG | $\mathbf{0.411}_{\pm0.004}$ | $\mathbf{0.265}_{\pm0.002}$ | $\mathbf{0.183}_{\pm0.002}$ | $\mathbf{0.132}_{\pm0.003}$ | $\mathbf{0.186}_{\pm0.002}$ | $\mathbf{0.299}_{\pm0.004}$ | 30.3% | $\mathbf{0.477}_{\pm0.004}$ | $\mathbf{0.484}_{\pm0.004}$ | $\mathbf{0.480}_{\pm0.004}$ |

aforementioned obtained labels, and use the computed scores as results of the CE metrics.

In addition to automatic evaluations, we also perform human evaluations of the quality of the generated reports. For each report, we ask three annotators with medical backgrounds to assess its quality in three aspects: fluency, completeness, and precision. The guideline is illustrated in Table 2. The annotators are asked to rate each aspect of the report on a scale of 1 to 3 accordingly, with higher scores indicating better quality. The quality of a report is measured by the average scores from different annotators.

***Implementation Details.*** For model architecture, we use the standard Transformer encoder with three multi-head attention layers as $f_{FE}$, and adopt two different eight-layer Transformers for $f_{VE}$ in TNG and $f_{DE}$, respectively. The number of the attention head and dimension of the hidden states for all modules are set to 8 and 512, respectively. For the diffusion networks, the total time step for diffusion forwarding and decoding processes $T$ is set to 20. We also follow the standard process of the denoising diffusion implicit model (DDIM) [44] sampler in the decoding process. For optimization, we use Adam optimizer to update all model parameters with a learning rate of $5e - 4$. We follow the learning rate scheduling strategy in Vaswani et al. [46] with $20,000$ steps for warm-up, and train the model on IU X-Ray and MIMIC-CXR with 300 and 10 epochs,

**Table 4: Human evaluation results on the report generated by different models using the sampled test instances from MIMIC-CXR. The range of human evaluation scores is from 1 to 3. "F", "C", and "P" denote fluency, completeness, and precision, respectively. "IAA" means the inter-annotator agreement (i.e., the number of scores agreed by all annotators out of all annotations).**

| DATA | MODEL | F | C | P | AVG. | IAA |
|---|---|---|---|---|---|---|
| IU X-RAY | TRANS | 2.6 | 2.1 | 2.0 | 2.2 | 82% |
| | DIFF | 2.7 | 2.2 | 2.0 | 2.3 | 80% |
| | DIFF+TNG | 2.8 | 2.4 | 2.1 | 2.4 | 82% |
| MIMIC-CXR | TRANS | 2.5 | 1.9 | 1.8 | 2.1 | 84% |
| | DIFF | 2.5 | 2.0 | 1.9 | 2.1 | 82% |
| | DIFF+TNG | 2.7 | 2.2 | 2.0 | 2.3 | 86% |

respectively. For other hyper-parameter settings, we try different combinations of them and select the one with the best performance on the validation set in our final experiments. For all experiments, we run them three times with different random seeds and report the average and standard deviation.

**Table 5: Performance comparison of our approach with the state-of-the-art studies on test sets of IU X-Ray and MIMIC-CXR with respect to NLG and CE metrics. The best results of different metrics are highlighted in boldface. For LLM-based approaches (i.e., XrayGPT), we illustrate the number of parameters in parentheses.**

| Data | Model | NLG Metrics | | | | | | CE Metrics | | |
|---|---|---|---|---|---|---|---|---|---|---|
| | | BL-1 | BL-2 | BL-3 | BL-4 | MTR | RG-L | P | R | F1 |
| IU X-Ray | ST [47] | 0.216 | 0.124 | 0.087 | 0.066 | - | 0.306 | - | - | - |
| | Att2in [41] | 0.224 | 0.129 | 0.089 | 0.068 | - | 0.308 | - | - | - |
| | AdaAtt [30] | 0.220 | 0.127 | 0.089 | 0.068 | - | 0.308 | - | - | - |
| | CoAtt [16] | 0.455 | 0.288 | 0.205 | 0.154 | - | 0.369 | - | - | - |
| | Hrgr [25] | 0.438 | 0.298 | 0.208 | 0.151 | - | 0.322 | - | - | - |
| | Cmas-RL [15] | 0.464 | 0.301 | 0.210 | 0.154 | - | 0.362 | - | - | - |
| | R2Gen [4] | 0.470 | 0.304 | 0.219 | 0.165 | - | 0.371 | - | - | - |
| | CA [29] | 0.492 | 0.314 | 0.222 | 0.169 | 0.193 | 0.381 | - | - | - |
| | CMCL [27] | 0.477 | 0.305 | 0.217 | 0.162 | 0.186 | 0.378 | - | - | - |
| | PPKED [28] | 0.483 | 0.315 | 0.224 | 0.168 | - | 0.376 | - | - | - |
| | R2GenCMN [3] | 0.475 | 0.309 | 0.222 | 0.170 | 0.191 | 0.375 | - | - | - |
| | R2GenRL [39] | 0.494 | 0.321 | 0.235 | 0.181 | 0.201 | 0.384 | - | - | - |
| | XrayGPT (7B) [37] | 0.177 | 0.104 | 0.047 | 0.007 | 0.105 | 0.203 | - | - | - |
| | **ControlDiff** | **0.508** | **0.332** | **0.243** | **0.189** | **0.207** | **0.390** | - | - | - |
| MIMIC -CXR | ST [47] | 0.299 | 0.184 | 0.121 | 0.084 | 0.124 | 0.263 | 0.249 | 0.203 | 0.204 |
| | Att2in [41] | 0.325 | 0.203 | 0.136 | 0.096 | 0.134 | 0.276 | 0.322 | 0.239 | 0.249 |
| | AdaAtt [30] | 0.299 | 0.185 | 0.124 | 0.088 | 0.118 | 0.266 | 0.268 | 0.186 | 0.181 |
| | Topdown [1] | 0.317 | 0.195 | 0.130 | 0.092 | 0.128 | 0.267 | 0.320 | 0.231 | 0.238 |
| | R2Gen [4] | 0.353 | 0.218 | 0.145 | 0.103 | 0.142 | 0.277 | 0.333 | 0.273 | 0.276 |
| | CA [29] | 0.350 | 0.219 | 0.152 | 0.109 | 0.151 | 0.283 | - | - | - |
| | CMCL [27] | 0.344 | 0.217 | 0.140 | 0.097 | 0.133 | 0.281 | - | - | - |
| | PPKED [28] | 0.360 | 0.224 | 0.149 | 0.106 | 0.149 | 0.284 | - | - | - |
| | R2GenCMN [3] | 0.353 | 0.218 | 0.148 | 0.106 | 0.142 | 0.278 | 0.334 | 0.275 | 0.278 |
| | R2GenRL [39] | 0.381 | 0.232 | 0.155 | 0.109 | 0.151 | 0.287 | 0.342 | 0.294 | 0.292 |
| | WarmStart [34] | 0.392 | 0.245 | 0.169 | 0.124 | 0.153 | 0.285 | 0.359 | 0.412 | 0.384 |
| | ITA [49] | 0.395 | 0.253 | 0.170 | 0.121 | 0.147 | 0.284 | - | - | - |
| | WarmStart [34] | 0.392 | 0.245 | 0.169 | 0.124 | 0.153 | 0.285 | 0.359 | 0.412 | 0.384 |
| | RGRG [45] | 0.373 | 0.249 | 0.175 | 0.126 | 0.168 | 0.264 | 0.461 | 0.475 | 0.447 |
| | ORGan [12] | 0.407 | 0.256 | 0.172 | 0.123 | 0.162 | 0.293 | 0.416 | 0.418 | 0.385 |
| | KiUT [13] | 0.393 | 0.243 | 0.159 | 0.113 | 0.160 | 0.285 | 0.371 | 0.318 | 0.321 |
| | XrayGPT (7B) [37] | 0.128 | 0.045 | 0.014 | 0.004 | 0.079 | 0.111 | - | - | - |
| | **ControlDiff** | **0.411** | **0.265** | **0.183** | **0.132** | **0.186** | **0.299** | **0.477** | **0.484** | **0.480** |

## 4.2 Overall Results

Experiment results of different models on the two benchmark datasets are reported in Table 3, with several observations drawn as follows. It is observed that the basic non-AR model ("Diff") consistently outperforms the AR one ("Trans") on both datasets, where the reason owes to that the error propagation problem is alleviated by diffusion networks through synchronous generation. On top of "Diff", our full model "Diff+TNG" obtains further improvements through leveraging task-specific noise rather than stochastic Gaussian noise in diffusion networks, confirming the effectiveness of noise control. The possible reason behind this observation is that the task-specific noise provides more precise hints to the diffusion process, therefore ensuring the quality of generated reports by eliminating potential irrelevant contents.

For human evaluation, we randomly sample 50 instances from the test sets of IU X-Ray and MIMIC-CXR and collect the reports generated by different models (i.e., "Trans", "Diff", "Diff+TNG"). The results of the human evaluation are reported in Table 4. We also report the inter-annotator agreement (IAA) that computes the number of scores agreed by all annotators out of all annotations. Similar to the trend in Table 3, human evaluation results show that our approach outperforms all baselines, which further confirms the effectiveness of our approach.

Moreover, we compare it with existing state-of-the-art solutions on both datasets, with results presented in Table 5. Overall, our approach outperforms other AR-based solutions on all metrics, illustrating the superiority of our approach for RRG. Notably, our approach even achieves better performance than the studies that based on large language models (LLMs) (i.e., XrayGPT), indicating

Table 6: Performance (i.e., the average and standard deviation of three runs with different random seeds) comparison of our approach under different settings of noise control on IU X-Ray and MIMIC-CXR with respect to NLG metrics. "GN" means the diffusion model with the standard stochastic Gaussian noise; "GVN" and "GTN" are global visual noise and global text noise, respectively; "LVN" and "LTN" are local visual noise and local text noise, respectively. "✓" means the noise type is used in the model; the last row where "GVN", "GTN", "LVN", and "LTN" are used is our full model.

(a) IU X-Ray

| ID | NOISE TYPE | | | | | EVALUATION METRIC | | | | | |
|----|----|----|----|----|----|----|----|----|----|----|----|
| | GN | GVN | GTN | LVN | LTN | BL-1 | BL-2 | BL-3 | BL-4 | MTR | RG-L |
| 1 | ✓ | | | | | $0.414_{\pm0.004}$ | $0.245_{\pm0.003}$ | $0.162_{\pm0.004}$ | $0.109_{\pm0.002}$ | $0.162_{\pm0.003}$ | $0.312_{\pm0.002}$ |
| 2 | | ✓ | | | | $0.445_{\pm0.003}$ | $0.267_{\pm0.003}$ | $0.188_{\pm0.002}$ | $0.130_{\pm0.004}$ | $0.168_{\pm0.003}$ | $0.342_{\pm0.002}$ |
| 3 | | | ✓ | | | $0.442_{\pm0.002}$ | $0.265_{\pm0.002}$ | $0.185_{\pm0.004}$ | $0.132_{\pm0.004}$ | $0.166_{\pm0.002}$ | $0.340_{\pm0.003}$ |
| 4 | | | | ✓ | | $0.443_{\pm0.002}$ | $0.264_{\pm0.003}$ | $0.189_{\pm0.004}$ | $0.134_{\pm0.002}$ | $0.167_{\pm0.002}$ | $0.341_{\pm0.003}$ |
| 5 | | | | | ✓ | $0.447_{\pm0.002}$ | $0.268_{\pm0.002}$ | $0.190_{\pm0.003}$ | $0.134_{\pm0.004}$ | $0.170_{\pm0.003}$ | $0.344_{\pm0.003}$ |
| 6 | | ✓ | ✓ | | | $0.458_{\pm0.002}$ | $0.308_{\pm0.003}$ | $0.216_{\pm0.002}$ | $0.156_{\pm0.004}$ | $0.189_{\pm0.003}$ | $0.367_{\pm0.002}$ |
| 7 | | | | ✓ | ✓ | $0.460_{\pm0.003}$ | $0.305_{\pm0.004}$ | $0.218_{\pm0.002}$ | $0.159_{\pm0.003}$ | $0.188_{\pm0.003}$ | $0.369_{\pm0.003}$ |
| 8 | | ✓ | | ✓ | | $0.463_{\pm0.003}$ | $0.313_{\pm0.002}$ | $0.217_{\pm0.003}$ | $0.158_{\pm0.002}$ | $0.186_{\pm0.003}$ | $0.371_{\pm0.004}$ |
| 9 | | | ✓ | | ✓ | $0.462_{\pm0.003}$ | $0.311_{\pm0.002}$ | $0.216_{\pm0.002}$ | $0.157_{\pm0.002}$ | $0.187_{\pm0.003}$ | $0.370_{\pm0.004}$ |
| 10 | ✓ | ✓ | ✓ | ✓ | ✓ | $0.505_{\pm0.002}$ | $0.330_{\pm0.003}$ | $0.242_{\pm0.004}$ | $0.186_{\pm0.002}$ | $0.206_{\pm0.002}$ | $\mathbf{0.391}_{\pm0.002}$ |
| 11 | | ✓ | ✓ | ✓ | ✓ | $\mathbf{0.508}_{\pm0.003}$ | $\mathbf{0.332}_{\pm0.002}$ | $\mathbf{0.243}_{\pm0.003}$ | $\mathbf{0.189}_{\pm0.002}$ | $\mathbf{0.207}_{\pm0.002}$ | $0.390_{\pm0.003}$ |

(b) MIMIC-CXR

| ID | NOISE TYPE | | | | | EVALUATION METRIC | | | | | |
|----|----|----|----|----|----|----|----|----|----|----|----|
| | GN | GVN | GTN | LVN | LTN | BL-1 | BL-2 | BL-3 | BL-4 | MTR | RG-L |
| 1 | ✓ | | | | | $0.373_{\pm0.002}$ | $0.217_{\pm0.003}$ | $0.142_{\pm0.002}$ | $0.101_{\pm0.004}$ | $0.134_{\pm0.002}$ | $0.274_{\pm0.003}$ |
| 2 | | ✓ | | | | $0.387_{\pm0.003}$ | $0.229_{\pm0.002}$ | $0.156_{\pm0.004}$ | $0.114_{\pm0.003}$ | $0.151_{\pm0.002}$ | $0.279_{\pm0.003}$ |
| 3 | | | ✓ | | | $0.385_{\pm0.003}$ | $0.230_{\pm0.004}$ | $0.153_{\pm0.002}$ | $0.109_{\pm0.002}$ | $0.148_{\pm0.004}$ | $0.280_{\pm0.003}$ |
| 4 | | | | ✓ | | $0.383_{\pm0.002}$ | $0.234_{\pm0.003}$ | $0.157_{\pm0.003}$ | $0.110_{\pm0.004}$ | $0.146_{\pm0.003}$ | $0.281_{\pm0.002}$ |
| 5 | | | | | ✓ | $0.384_{\pm0.003}$ | $0.232_{\pm0.004}$ | $0.155_{\pm0.002}$ | $0.111_{\pm0.003}$ | $0.150_{\pm0.004}$ | $0.283_{\pm0.002}$ |
| 6 | | ✓ | ✓ | | | $0.397_{\pm0.004}$ | $0.251_{\pm0.004}$ | $0.168_{\pm0.002}$ | $0.126_{\pm0.003}$ | $0.169_{\pm0.002}$ | $0.285_{\pm0.003}$ |
| 7 | | | | ✓ | ✓ | $0.391_{\pm0.003}$ | $0.250_{\pm0.004}$ | $0.170_{\pm0.002}$ | $0.124_{\pm0.003}$ | $0.168_{\pm0.002}$ | $0.289_{\pm0.003}$ |
| 8 | | ✓ | | ✓ | | $0.401_{\pm0.003}$ | $0.254_{\pm0.004}$ | $0.172_{\pm0.003}$ | $0.129_{\pm0.004}$ | $0.172_{\pm0.003}$ | $0.287_{\pm0.003}$ |
| 9 | | | ✓ | | ✓ | $0.395_{\pm0.004}$ | $0.253_{\pm0.003}$ | $0.174_{\pm0.003}$ | $0.126_{\pm0.004}$ | $0.171_{\pm0.002}$ | $0.292_{\pm0.003}$ |
| 10 | ✓ | ✓ | ✓ | ✓ | ✓ | $0.408_{\pm0.002}$ | $0.262_{\pm0.003}$ | $0.180_{\pm0.002}$ | $0.132_{\pm0.004}$ | $\mathbf{0.187}_{\pm0.002}$ | $\mathbf{0.302}_{\pm0.003}$ |
| 11 | | ✓ | ✓ | ✓ | ✓ | $\mathbf{0.411}_{\pm0.004}$ | $\mathbf{0.265}_{\pm0.002}$ | $\mathbf{0.183}_{\pm0.002}$ | $\mathbf{0.132}_{\pm0.003}$ | $0.186_{\pm0.002}$ | $0.299_{\pm0.004}$ |

that appropriate modeling of the report generation process is more efficient than using massive parameters in LLMs.

## 4.3 Effect of Different Noise Types

To explore the impact of controlling different noise types, we run experiments on using particular task-specific information as noise in diffusion networks. Table 6 reports the results on two benchmark datasets, where "GN" refers to Gaussian noise; "GVN" and "GTN" represent the global visual and textual noise, respectively; "LGN" and "LTN" denote the local visual and textual noise, respectively. Several observations from different perspectives are illustrated as follows. First, compared with the model with the standard Gaussian noise (i.e., ID=1), our approach (ID=2-11) with any type of task-specific noise achieves better performance on most evaluation metrics. This observation demonstrates the effectiveness of using task-related noise in diffusion models for improving model performance. Second, comparing our approach with global noise

(ID=6) and with local noise (ID=7), the models achieve similar performance, showing both global and local noise contribute to the task; similarly, our approaches with visual (ID=8) and text noise (ID=9) obtain similar performance, showing both visual and text features are essential to the task. Third, we observe that models with multi-modal features as noise (ID=6-9) obtain improvements over the ones using single-modal features (ID=2-5), since controlling visual- or textual-only noise provides coarse guidance for report generation. Finally, we observe that our approaches with (ID=10) and without (ID=11) the standard Gaussian noise achieve comparable performance. This observation is intuitive since the Gaussian noise is task-irrelevant and thus brings limited useful information compared with other task-specific types of noise. Furthermore, our approach obtains the best performance, where leveraging various noise information from different views and modalities refines the iterative generation process.

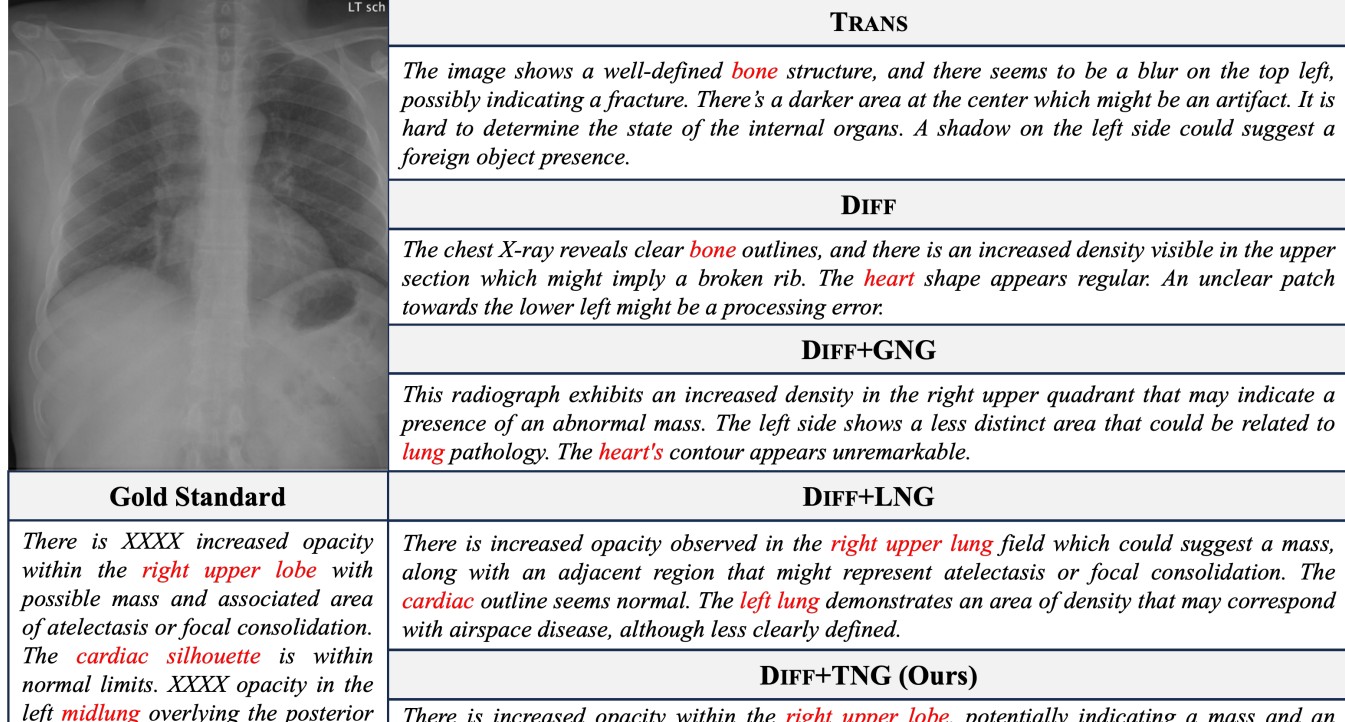

**Figure 2: An illustration of the report generation processes (through texts generated at different time steps) by different models with an example radiograph. Medical terms shared by the model outputs and the gold standard texts are highlighted in the same color. "GNG" and "LNG" stand for global noise generator and local noise generator, respectively.**

## 4.4 Case Study

To further qualitatively demonstrate the effect of noise control in our approach, we present a case study in Figure 2 with an example input radiograph selected from the test set of MIMIC-CXR, as well as the reports generated by "Trans", "Diff", "Diff+GNG", "Diff+LNG" and "Diff+TNG", where "Diff+GNG" and "Diff+LNG" refer to the diffusion networks equipped with only GNG or LNG, respectively, and medical terms shared by model output and gold standard are highlighted in same colors. It is observed that "Diff" generates reports with more medical terms related to the input radiograph than "Trans" since the error propagation problem of AR models is alleviated by "Diff". Meanwhile, the reports generated by "Diff" still contain irrelevant descriptions, since a less controlled generation process is performed owning to the use of stochastic Gaussian noise. "Diff+GNG" and "Diff+LNG" improve the quality of generated reports compared to "Diff". "Diff+GNG" effectively eliminates irrelevant descriptions by controlling global noises; "Diff+LNG" offers more fine-grained noise control for the diffusion networks and produces reports with more related medical terms than "Diff+GNG". Finally, "Diff+TNG" obtains the most elaborated reports compared to all aforementioned models, suggesting

that controlling both global and local noise provides comprehensive information for diffusion networks to generate precise reports.

## 5 CONCLUSION

In this paper, we propose ControlDiff that utilizes diffusion networks to generate the report for RRG and thus does not suffer from the error propagation issues of the existing approaches that use AR models. In addition, we enhance the diffusion networks with task-specific noise (e.g., global and local visual and text features) rather than the standard stochastic Gaussian noise used in the standard diffusion networks, to generate precise reports for RRG. Experimental results on two widely used English benchmark RRG datasets, namely, IU X-Ray and MIMIC-CXR, indicate the superiority and effectiveness of our proposed approach compared to existing studies, where our approach outperforms strong baselines and existing studies on both datasets. Further analyses and case study explore the effect of our noise controlling mechanisms from different perspectives, suggesting that our approach presents its potential of being a reference framework to conduct a controlled generation process for other related tasks in future studies.

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
