# OpenReview forum: "Diffusion Networks with Task-Specific Noise Control for Radiology Report Generation"
_acmmm.org/ACMMM/2024/Conference — MM2024 Poster_

### Official Review · Reviewer_UcVd · 2024-05-16

**Rating:** 4
**Confidence:** 3

**Summary:**

Overall, the paper is well written and proposes a novel method to improve over standard autoregressive models using diffusion models by employing a prior distribution to improve over baselines in literature.

**Strengths:**

- Task-specific noise is very intuitive since one can think of this as introducing a prior over the initial noise distribution to better model the underlying data distribution
- Adding global and local noise is intuitive since each module handles a different kind of information (specific to the dataset vs specific to each instance)
- Good work on the experiment section. Detailed ablations for the different noise vectors and their combinations in terms of multimodality, global vs local etc show the importance of each module in a clear and consistent manner

**Limitations:**

There is a considerable amount of missing information about certain hyperparameters and the models used to create the noise vector
- How do you determine the top N n-grams used for the GNG module?
- Since the two datasets consist of chest reports only, these radiology reports usually uniformly talk about all the pathologies present in the study.  The frequent n grams used as noise in the global text module could be a very strong starting point for the model as they might just turn out to be the crucial concepts which are ideally intended to be detected by the image model. Perhaps a word cloud in the appendix might help remove this suspicion. This would naturally not be a problem for a dataset consisting of reports for multiple body parts. - If this turns out to be an observation, please suggest a method to remove this bias to avoid information leakage.
- What is the word embedding module used to map the ngrams to their respective representations?
- What is the intuition behind sampling a noise vector in the global and local text modules over say taking the mean? If this is done separately for each token, it makes more sense, however, the authors haven’t clearly indicated this.
- How do you construct n_GV from the background feature vector for the entire training set exactly? I assume you would have a separate vector for each image considered.
- I reckon the embedding spaces of the n grams and radiograph background might not be in the same space since its not from a pre-trained model like CLIP. How did you manage this? The same question would hold for the local module as well.
- It is not very clear how the authors trained a transformer-based model to annotate medical terms used in constructing the local noise vector. Any information about the implementation would throw some light here.
- The authors propose averaging the text and the vision vector for the global noise vector but not the local noise vector. What is the intuition behind this change?
- Although the baseline involving Diff and Diff + TNG shows how TNG can help improve over Diff. The baseline Trans looks to be underpowered compared to the other baselines under consideration. DDCap uses a VIT-B for its visual encoder and GPT-2 small for their decoder which is much bigger than the architecture described for Trans. Please make sure all the baselines	A match exactly in terms of the model architecture to help validate the improvements of Diffusion models over autoregressive baselines.
- Although the ablations are very detailed, overlap-based metrics are not really informative when it comes to language modelling, adding Cross entropy-based metrics for the MIMIC-CXR dataset in Table 6 would improve and strengthen this ablation study.

Suggestions
- It is very interesting to note that the Inter annotator agreement for the stochastic noise-based diffusion models was lower than the Autoregressive models. Do you know why this was the case or can you hypothesize?

Here are some minor format errors
- Lines 183 and 186 mention LNC instead of LNG. Same for GNC in lines 186 and 225
- Warmstart has been added twice to the Mimic-CXR subset in Table 5
- The variables in lines 352 and 397 have typos.

**Suitability:**

3

---

### Official Review · Reviewer_aXxq · 2024-05-23

**Rating:** 5
**Confidence:** 4

**Summary:**

This paper introduces ControlDiff, a non-autoregressive (non-AR) solution for Radiology Report Generation (RRG) using diffusion networks with a task-specific noise control mechanism. The proposed method aims to enhance the generation process by appropriately handling essential cross-modal information through noise control strategies. The paper addresses an important problem in medical imaging with a unique method. However, additional details on implementation and theoretical aspects would strengthen the paper's contribution to the field.

**Strengths:**

1. Innovative Approach: The introduction of a non-AR solution for RRG using diffusion networks with task-specific noise control is a novel contribution.
2. Experimental Evaluation: The paper provides experimental results on benchmark datasets demonstrating the superiority of the proposed approach against state-of-the-art methods.
3. Clear Problem Statement: The paper effectively highlights the challenges in existing AR-based approaches for RRG and proposes a targeted solution with clear explanations of the methodology.

**Limitations:**

1. Lack of Detailed Implementation: The paper lacks in-depth details of the implementation of the ControlDiff model, which could hinder reproducibility.
2. Limited Comparative Analysis: While the paper claims superiority over state-of-the-art studies, a more extensive comparison with other relevant methods in the field would strengthen the evaluation.
3. Lack of Theoretical Analysis: The paper could benefit from a more detailed theoretical analysis of the diffusion model and noise control mechanism to provide a deeper understanding of the proposed approach.

**Suitability:**

3

---

### Official Review · Reviewer_5jhf · 2024-05-27

**Rating:** 4
**Confidence:** 3

**Summary:**

This paper proposed a new framework for Radiology Report Generation with a diffusion model, in which medical task-specific noise control was added locally and globally.

**Strengths:**

Overall, the paper is clearly written, and the main and ablation experiments are well conducted.

ControlDiff departs from the traditional random Gaussian noise and uses medical content that is more compatible with the Radiology Report for noise generation, making the model itself more suitable for the Radiology Report. In my opinion, this kind of noise scheduler is innovative for both the study of diffusion models and the Radiology Report Generation.

From the experiments, the comparisons are adequate and the results of ControlDiff are impressive, surpassing previous methods on all evaluation metrics. It has the opportunity to become a new baseline in this field of research.

**Limitations:**

From a diffusion model perspective, random Gaussian noise is indeed information-independent, so much research has been devoted to making text or image representations capable of guiding the denoising process by adding cross-attention layers. Could ControlDiff's medical domain-specific noise addition be switched to cross-attention-guided denoising?

The design of global and local noise for both modalities is somewhat complex, and only the presentation of evaluation metrics is not enough; more case studies could highlight the role of each small component.

I'm not familiar with the relevant literature in the field. Since the authors don't mention it, I am curious if ControlDiff was the first to use diffusion models in the field of Radiology Report Generation. If not, are there any previous studies that need to be compared?

**Suitability:**

2

---

### Meta-Review · Area_Chair_4QGL · 2024-06-29

**Recommendation:** Accept (Poster)
**Confidence:** 5

**Metareview:**

The objective of the method shows promise, yet the experiment section requires more ablations for more robust validation. Key limitations include missing details on hyperparameters and models used for the noise vector creation and ambiguities regarding the selection and handling of n-grams in the GNG module. The authors are suggested to address all these comments (including other limitations highlighted by the reviewers) in the camera-ready version to enhance the paper's clarity and rigor.